# Bird migratory flyways influence the phylogeography of the invasive brine shrimp *Artemia franciscana* in its native American range

Joaquín Muñoz[1], Francisco Amat[2], Andy J. Green[1], Jordi Figuerola[1] and Africa Gómez[3]

[1] Department of Wetland Ecology, Estación Biológica de Doñana (CSIC), Seville, Spain
[2] Instituto de Acuicultura de Torre de la Sal (CSIC), Castellón, Spain
[3] Department of Biological Sciences, University of Hull, Hull, UK

Corresponding author
Africa Gómez, a.gomez@hull.ac.uk

## ABSTRACT

Since Darwin's time, waterbirds have been considered an important vector for the dispersal of continental aquatic invertebrates. Bird movements have facilitated the worldwide invasion of the American brine shrimp *Artemia franciscana*, transporting cysts (diapausing eggs), and favouring rapid range expansions from introduction sites. Here we address the impact of bird migratory flyways on the population genetic structure and phylogeography of *A. franciscana* in its native range in the Americas. We examined sequence variation for two mitochondrial gene fragments (COI and 16S for a subset of the data) in a large set of population samples representing the entire native range of *A. franciscana*. Furthermore, we performed Mantel tests and redundancy analyses (RDA) to test the role of flyways, geography and human introductions on the phylogeography and population genetic structure at a continental scale. *A. franciscana* mitochondrial DNA was very diverse, with two main clades, largely corresponding to Pacific and Atlantic populations, mirroring American bird flyways. There was a high degree of regional endemism, with populations subdivided into at least 12 divergent, geographically restricted and largely allopatric mitochondrial lineages, and high levels of population structure ($\Phi_{ST}$ of 0.92), indicating low ongoing gene flow. We found evidence of human-mediated introductions in nine out of 39 populations analysed. Once these populations were removed, Mantel tests revealed a strong association between genetic variation and geographic distance (i.e., isolation-by-distance pattern). RDA showed that shared bird flyways explained around 20% of the variance in genetic distance between populations and this was highly significant, once geographic distance was controlled for. The variance explained increased to 30% when the factor human introduction was included in the model. Our findings suggest that bird-mediated transport of brine shrimp propagules does not result in substantial ongoing gene flow; instead, it had a significant historical role on the current species phylogeography, facilitating the colonisation of new aquatic environments as they become available along their main migratory flyways.

## INTRODUCTION

Since *Darwin (1859)*, the role of birds as dispersal vectors for the diapausing propagules of continental aquatic organisms has been recognized (*Carlquist, 1983*; *Bilton, Freeland & Okamura, 2001*; *Green & Figuerola, 2005*). A range of studies have emphasized the importance of dispersal at short and long range by waterbirds for both passively dispersed aquatic invertebrates – through their diapausing eggs or other dispersing stages – and plants – through their seeds – (*Figuerola & Green, 2002*; *Green & Figuerola, 2005*; *Green et al., 2008*; *Brochet et al., 2010a*; *Brochet et al., 2010b*; *van Leeuwen et al., 2012a*; *van Leeuwen et al., 2012b*). Such long-distance, bird-mediated dispersal between aquatic habitats should result in high population gene flow and reduced population or phylogeographic structure. In stark contrast to this prediction, ongoing gene flow between populations has consistently been found to be low (*Boileau, Hebert & Schwartz, 1992*; *De Meester et al., 2002*) and phylogeographic structures quite marked, with high levels of endemism (*Gómez, Carvalho & Lunt, 2000*; *Gómez et al., 2007a*; *De Gelas & De Meester, 2005*; *Muñoz et al., 2008*). This paradox has been explained through a combination of high population growth rates, rapid local adaptation and a buffering effect of large egg banks accumulated in sediments, resulting in a monopolisation of resources by the few initial founders, reducing the impact of further immigrants on population structure – what was termed the "monopolisation hypothesis" (*De Meester et al., 2002*). Consistent with this hypothesis, several studies failed to uncover any relationship between the geographic distribution of genetic lineages and bird migration patterns (*Gómez et al., 2007a*; *Mills, Lunt & Gómez, 2007*; *Muñoz et al., 2008*). In contrast, the perceived similarity between bird migratory pathways and the distribution of passively dispersed invertebrate genetic lineages suggests that waterfowl are important dispersal vectors (*Taylor, Finston & Hebert, 1998*; *Freeland, Romualdi & Okamura, 2000*; *Hebert, Witt & Adamowicz, 2003*). In fact, *Figuerola, Green & Michot (2005)* tested explicitly the relationship between bird movements and aquatic invertebrate population genetic structure, revealing a significant association between historical ringing data – used as a proxy of bird-mediated dispersal between populations – and population genetic distances for two cladocerans and a bryozoan in North America, concluding that birds significantly contributed to effective dispersal.

Given that continental aquatic invertebrates are unlikely to be in migration-drift equilibrium (*Boileau, Hebert & Schwartz, 1992*; *Gómez et al., 2007a*), recent studies have interpreted population isolation-by-distance (IBD) patterns as a signature of historical patterns due to sequential colonisation events, as newly available habitats are more likely to be colonised by nearby populations, with little further impact of gene flow (*Gómez et al., 2007a*; *Mills, Lunt & Gómez, 2007*; *Muñoz et al., 2008*). Therefore, associations between bird movements and genetic distance in aquatic invertebrates based on mitochondrial markers

could result from bird-mediated historical colonisation of newly available habitats, instead of ongoing gene flow (*Figuerola, Green & Michot, 2005*). Shedding light on the role of bird movements on the geographic distribution of genetic lineages would help us to understand the structuring of genetic diversity and phylogeography in passively dispersed aquatic invertebrates.

*Artemia franciscana* (Kellogg, 1906)(Crustacea: Anostraca), the most widely distributed brine shrimp in America, occurs in hypersaline habitats from Canada to Chile and many Atlantic islands (*Hontoria & Amat, 1992*). It is found in a wide diversity of isolated water bodies, including coastal rock pools and lagoons, inland playas and high mountain salt lakes, permanent prairie salt lakes and commercial salt works (*Van Stappen, 2002*), spanning an extreme range of water chemistry compositions and salinity from high carbonate, or high sulphate athalasic waters to seawater salterns (*Bowen, Buoncristiani & Carl, 1988*). It is a sexual species, and females produce two types of eggs: subitaneous eggs in benign environmental conditions suitable for population growth, and diapausing eggs (i.e., cysts) during adverse conditions. *Artemia* cysts are amongst the most resistant animal life forms, surviving extreme environmental stresses including UV radiation, desiccation, thermal extremes and anoxia (*Clegg, 2005*). Cysts accumulate at the shoreline and in egg banks in lake sediments (*Moscatello & Belmonte, 2009*), and are readily dispersed by birds, which are the main vectors between catchments. Wind dispersal occurs but over much shorter distances (<1 km, *Vanschoenwinkel et al., 2009*). Many migratory bird species, especially shorebirds, use *Artemia* habitats and adult brine shrimp – often carrying viable cysts – can make up a substantial component of their diet (*Anderson, 1970*; *Sánchez, Green & Castellanos, 2005*; *Varo et al., 2011*; *Vest & Conover, 2011*). Birds can disperse cysts between habitats either externally – attached to their feathers or feet – or internally in their digestive tract (*Brochet et al., 2010b*; *Green et al., 2005*; *Green et al., 2013*; *Sánchez et al., 2007*; *Sánchez et al., 2012*). Research showing the internal transport of viable *A. franciscana* cysts in the field by the American Avocet, *Recurvirostra americana* (AJG, unpublished data), confirms shorebirds as an effective agent of dispersal in North America (see also *Green et al., 2005*). Recently, *Viana et al. (2013)* estimated the maximum dispersal for *Artemia* cysts via wildfowl as between 230 and 1209 km based on gut passage times of cysts ingested by captive ducks and the distances moved by wild ducks.

Populations of *A. franciscana* have substantial levels of genetic (*Abreu-Grobois & Beardmore, 1982*; *Gajardo et al., 1995*; *Maniatsi et al., 2009*) and morphological variation (*Hontoria & Amat, 1992*), and are locally adapted to the ionic composition of their habitats (*Bowen, Buoncristiani & Carl, 1988*). Indeed, effective reproductive isolation between some populations is due to different ranges of tolerance to ionic compositions (*Bowen, Buoncristiani & Carl, 1988*), and so this taxon is regarded by some authors as a "superspecies" (*Bowen, Buoncristiani & Carl, 1988*). Nevertheless, despite half a century of research for aquaculture and ecotoxicology, comprehensive large-scale phylogeographic surveys of *A. franciscana* are lacking.

Cysts from *A. franciscana* – harvested mainly from populations in the San Francisco Bay saltworks and the Great Salt Lake in the USA – have been used globally as a food

source in aquaculture and in the pet fish trade for decades (*Abatzopoulos, 2002*; *Amat et al., 2005*; *Amat et al., 2007*). Effluents from fish farms are likely to contain cysts that can potentially colonise nearby natural wetlands. In addition, the introduction of *A. franciscana* has been and still is promoted worldwide to increase salt production or to generate local sources of cysts until as recently as 1993 (*Tackaert & Sorgeloos, 1993*; *Sui et al., 2012*). As a result of such accidental and intentional inoculations, *A. franciscana* has become an invasive species in saline and hypersaline wetlands worldwide (*Muñoz & Pacios, 2010*; J Muñoz, A Gómez, J Figuerola, F Amat, C Rico, AJ Green, 2013 unpublished). For instance, this invasion has led to rapid local extinction of native *Artemia* species in the Mediterranean region (*Amat et al., 2005*). Commercial strains of *A. franciscana* were also introduced in various American sites in the 1970s (*Camara, 2001*; *Amat et al., 2004*). In Brazil, further spreading of the species, probably via bird movements, was noticed within a few years of its introduction in areas where it was previously absent (*Camara, 2001*). However, the impact of these introductions on the genetic diversity and structure of native American populations has yet to be investigated.

*Artemia franciscana* represents a very interesting model to test the effect of bird movements on the geographic distribution of genetic lineages and patterns of genetic variation in aquatic invertebrates since (1) its distribution encompasses three continental-scale bird migratory flyways spanning both North and South America (i.e., the Pacific, Central and Atlantic flyways), but is highly fragmented due to its habitat requirements (hypersaline lakes), (2) its habitats are frequented by migratory shorebirds; *Artemia* is an important prey of these and other waterbirds and its cysts can be readily quantified in waterbird excreta (*Green et al., 2005*; *Sánchez et al., 2007*), and (3) the intentional or accidental inoculations outside the native range may be affecting its natural population genetic structure.

Here, we carry out the first comprehensive phylogeographic study of *A. franciscana* throughout its known native range (i.e., from Central Canada to southern Chile and Argentina, including the Caribbean islands) using sequence variation for two mitochondrial genes (COI and 16S). Our results indicate a high level of genetic structure and endemism at a continental scale, identify the impact of human introductions and suggest a direct link between bird migratory routes (i.e., flyways) and the historical colonization of *A. franciscana* throughout the Americas, revealing a key role for birds in initial founder events.

## METHODS

### Samples, laboratory procedures, and sequences

We obtained samples from 39 *A. franciscana* populations across its American geographical distribution, from Canada to Chile and Argentina, including Caribbean islands and a population from Cape Verde (Table 1 and Fig. 1). Most samples were cysts obtained from the 'cyst-bank' of the Instituto de Acuicultura de Torre de la Sal (CSIC, Castellón, Spain), collected between 1984 and 2000. Four Canadian samples were collected in the field in 2009, two of them (MANW and CHAP) as adults, which were preserved in absolute ethanol until needed. An additional cyst sample from Mono Lake (USA), collected in the

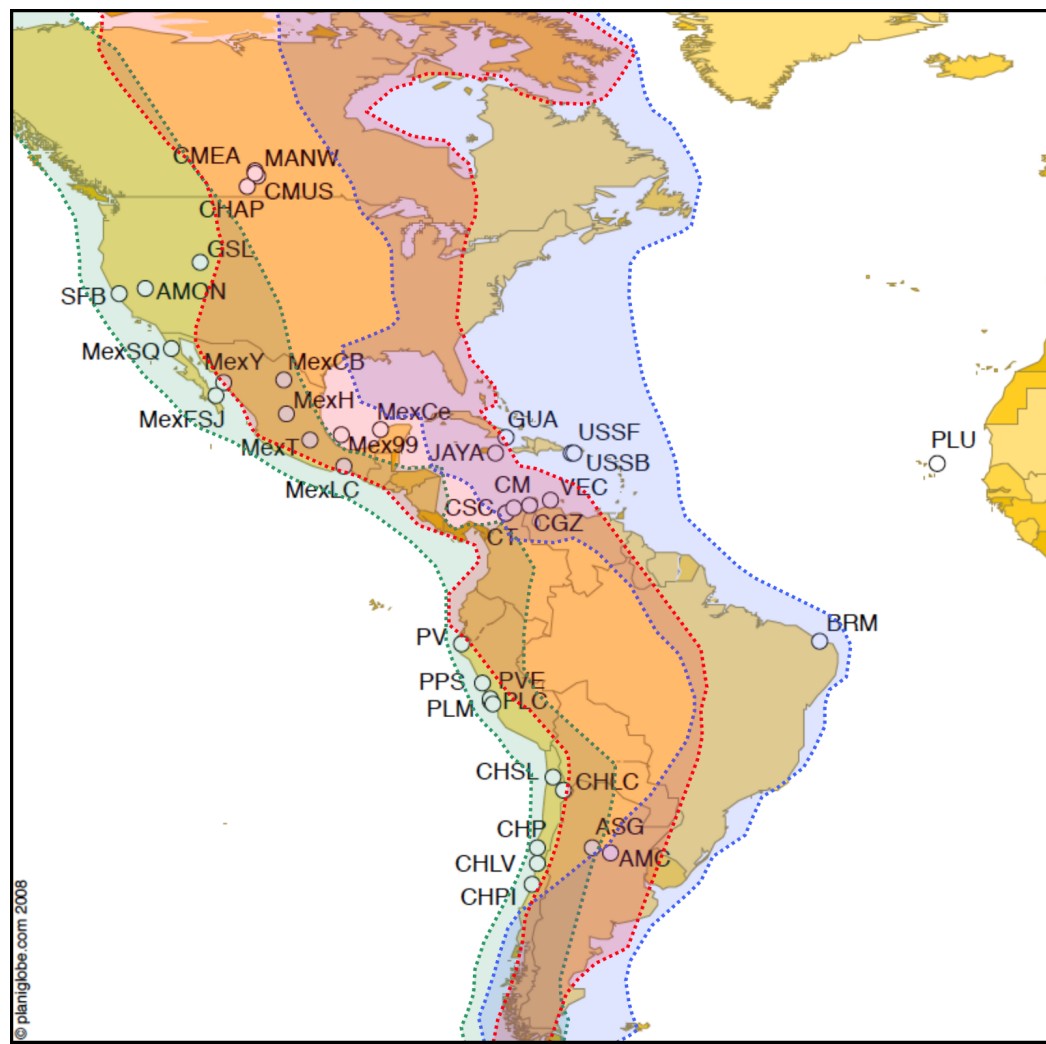

**Figure 1 Map of *Artemia franciscana* sampled sites and American bird migratory flyways.** The sampled populations are shown, with indication of the main American migratory flyways following Birdlife International (see text for details). Green shading: Pacific flyway, red shading: Central flyway, blue shading, Atlantic flyway.

1970s was kindly provided by the Artemia Reference Centre (ARC 270). A few cyst samples that yielded poor quality DNA extractions were subject to hatching and the resulting nauplii used for DNA extractions (i.e., MexCB and GUA samples).

DNA extractions were carried out on individual cysts (previously rinsed in distilled water), whole nauplii or partial adults using a HotSHOT protocol (*Montero-Pau, Gómez & Muñoz, 2008*). We used *Artemia*-specific primers 1/2COI_Fol-F and 1/2COI_Fol-R (*Muñoz et al., 2008*) to amplify and sequence a 709 bp fragment of the mitochondrial Cytochrome *c* Oxidase Subunit I gene (COI). We also amplified and sequenced a 535 bp fragment of the 16S ribosomal RNA gene for a subset of individuals carrying different COI haplotypes to facilitate comparison with other published sequences using primers 16Sar-5′/16Sbr-3′ (*Palumbi, 1996*). PCRs were performed in 20 µL total volume containing

Table 1 *Artemia franciscana* **populations sampled.** Population code, year of sampling, COI haplotype code, number of individuals per haplotype per population ($N_{HAP}$) and number of individuals sequenced per population ($N_{TOTAL}$), are given for each sampling site. $Hs$, Standard gene diversity; $\pi$, Nucleotide diversity; N.A., Insufficient data to calculate $Hs$ (the minimum sample size per population performed by RAREFAC was 11). The main commercialised USA populations (i.e., SFB and GSL) and haplotypes from these populations shared with other populations are shown in red. Populations considered as introduced are indicated in bold. Only populations collected as part of this study are included.

| Code | Population (year) | COI haplotypes | $N_{HAP}$ | $N_{TOTAL}$ | $Hs$ | $\pi$ | Refs. for introduction |
|------|-------------------|----------------|-----------|-------------|------|-------|------------------------|
| MexCe | Celestún, Yucatán, México (1984) | Af01 | 15 | 15 | 0.000 | 0.0000 | - |
| Mex99 | Real de las Salinas, Campeche, México (1999) | Af01 | 14 | 14 | 0.000 | 0.0000 | - |
| MexY | Yavaros, Sonora, México (1993) | Af02; Af03 | 1; 11 | 12 | 0.167 | 0.0003 | - |
| MexH | Salinas de Hidalgo, San Luis Potosí, México (1989) | Af04; Af05; Af06; Af07; Af08; Af09 | 1; 1; 1; 2; 3; 3 | 11 | 0.873 | 0.0091 | - |
| **MexT** | Texcoco, Estado de México, México (1989) | Af07; Af08; Af09; Af10; Af11 | 1; 2; 7; 1; 2 | 13 | 0.705 | 0.0099 | Introduced from SFB in 1975 (*Castro, 1993*) cited in *Castro et al. (2006)* |
| MexCB | Salinas Casa Blanca, Cuatro Ciénagas de Carranza, Coahuila, México (1995) | Af12; Af13; Af14; Af15; Af16; Af17 | 2; 6; 1; 3; 3; 1 | 16 | 0.817 | 0.0023 | - |
| **MexLC** | La Colorada lagoon, Oaxaca, México (1993) | Af18; Af19; Af20; Af21 | 3; 1; 8; 2 | 14 | 0.648 | 0.0041 | - |
| **MexSQ** | San Quintín, Baja California, México (??) | Af10; Af22; Af23; | 12; 1; 1 | 14 | 0.275 | 0.0005 | - |
| **MexFSJ** | Faro San José, Baja California, México (1991) | Af01; Af10; Af24 | 1; 2; 1 | 4 | N.A. | 0.0226 | - |
| **GUA** | Frank País, Guantánamo, Cuba (1994) | Af10; Af20; Af25 | 6; 1; 9 | 16 | 0.575 | 0.0019 | Introduced in the 70's in 7 saltworks, (*Gelabert & Solis, 1994*) cited in *Tizol-Correa (2009)*. |
| GSL | Great Salt Lake, Utah, USA (??) | Af10; Af18; Af20; Af21; Af26; Af27 | 1; 2; 21; 2; 2; 1 | 29 | 0.475 | 0.0028 | - |
| SFB | San Francisco Bay, California, USA (??) | Af10; Af18; Af20; Af25 | 26; 6; 4; 1 | 37 | 0.480 | 0.0033 | - |
| USSF | Salina Fraternidad, Puerto Rico, CaboRojo, USA (2000) | Af28; Af29 | 12; 4 | 16 | 0.400 | 0.0007 | - |
| USSB | Laguna de las Salinas Bastoncillo, Lajas, Puerto Rico, USA (2000) | Af28; Af30; Af31; Af32; Af33; Af34 | 6; 4; 1; 1; 1; 1 | 14 | 0.769 | 0.0017 | - |
| MANW | Little Manitou Lake, Saskatchewan, Canada (2009) | Af35; Af36 | 8; 1 | 9 | N.A. | 0.0004 | - |

| Code | Population (year) | COI haplotypes | $N_{HAP}$ | $N_{TOTAL}$ | Hs | π | Refs. for introduction |
|------|-------------------|----------------|-----------|-------------|-----|---|------------------------|
| CMUS | Muskiki Lake, Saskatchewan, Canada (2009) | Af35; Af37; Af38 | 12; 1; 1 | 14 | 0.275 | 0.0005 | - |
| CHAP | Chaplin Lake, Saskatchewan, Canada (2009) | Af39; Af40; Af41; Af42; Af43 | 11; 1; 1; 2; 1 | 16 | 0.533 | 0.0027 | - |
| CMEA | Meacham Lake, Saskatchewan, Canada (2009) | Af35; Af44; Af45; Af46; Af47 | 26; 1; 1; 1; 1 | 30 | 0.253 | 0.0005 | - |
| **BRM** | Mossoro, Grossos, Brazil (1994) | Af10 | 11 | 11 | 0.000 | 0.0000 | Introduced in 1977 from SFB (*Camara, 2001*) |
| CGZ | Salinas de Galerazamba, Colombia (1985) | Af48; Af49; Af50 | 1; 8; 7 | 16 | 0.592 | 0.0017 | - |
| CM | Salinas de Manaure, Colombia (1999) | Af51; Af52 | 15; 1 | 16 | 0.125 | 0.0002 | - |
| CSC | Salina Cero, Colombia (1999) | Af49; Af50; Af53; Af54; Af55 | 8; 4; 1; 1; 1 | 15 | 0.676 | 0.0015 | - |
| CT | Tayrona, Colombia (1999) | Af56; Af57; Af58 | 9; 4; 2 | 15 | 0.590 | 0.0023 | - |
| PPS | Playa Salinas, Ancash, Perú (1995) | Af59; Af60; Af61; Af62; Af63 | 6; 2; 1; 3; 1 | 13 | 0.756 | 0.0023 | - |
| PLC | Los Chimus, Perú (1992) | Af59; Af64 | 14; 1 | 15 | 0.133 | 0.0002 | - |
| PV | Virrilla, Piura, Perú (1996) | Af65 | 16 | 16 | 0.000 | 0.0000 | - |
| PVe | Humedales de Ventanilla, Callao, Perú (1996) | Af59; Af61; Af66; Af67 | 8; 1; 6; 1 | 16 | 0.642 | 0.0020 | - |
| PLM | La Milagrosa, Chilca, Perú (1993) | Af68; Af69 | 14; 2 | 16 | 0.233 | 0.0004 | - |
| VEC | Salinas de Cumaraguas, Venezuela (1994) | Af51; Af70; Af71 | 10; 4; 1 | 15 | 0.514 | 0.0009 | - |
| **JAYA** | Yallahs Pond, Jamaica (1998) | Af18; Af19 | 15; 1 | 16 | 0.125 | 0.0002 | Known since 1992, morphology extremely similar to SFB (*Castro et al., 2000*) |
| CHSL | Salar de Llamará, Chile (1994-lab) | Af72; Af73; Af74; Af75 | 5; 1; 1; 2 | 9 | N.A. | 0.0014 | |
| **CHLC** | Laguna Cejas, Salar de Atacama, Chile (1995-lab) | Af18 | 16 | 16 | 0.000 | 0.0000 | *Maniatsi et al. (2009)* found different haplotypes, which were native. |
| CHLV | Los Vilos, Poza Palo Colorado, Chile (1997) | Af76; Af77 | 6; 10 | 16 | 0.500 | 0.0008 | |
| **CHPI** | Pichilemu Cahuil saltworks, Chile (??-lab) | Af18; Af78 | 8; 8 | 16 | 0.533 | 0.0103 | Reportedly introduced by artisanal workers (*Gajardo et al., 1998*) no details as to when. |

Table 1 (*continued*)

| Code | Population (year) | COI haplotypes | $N_{HAP}$ | $N_{TOTAL}$ | Hs | π | Refs. for introduction |
|------|-------------------|----------------|-----------|-------------|-----|---|------------------------|
| CHP | Poza Pampilla IV Region, Chile (1997) | Af79; Af80 | 14; 1 | 15 | 0.133 | 0.0048 | - |
| AMC | Mar Chiquita, Córdoba, Argentina (1997) | Af81 | 16 | 16 | 0.000 | 0.0000 | - |
| ASG | Salinas Grandes, Córdoba, Argentina (2000) | Af72; Af73; Af74; Af82; Af83; Af84; Af85 | 7; 1; 1; 1; 1; 1 | 13 | 0.731 | 0.0018 | - |
| AMON | Mono Lake (1970s, ARC270) | Af87; Af88; Af89; Af90; Af91; Af92 | 1; 1; 1; 7; 1; 1 | 12 | 0.682 | 0.0016 | - |
| PLU | Pedra de Lume, Sal Island, Cape Verde (??) | Af86; Af93 | 15; 1 | 16 | 0.125 | 0.0002 | - |

$1\times$ reaction buffer, 2.0 mM MgCl$_2$, 0.2 mM dNTPs, 0.6 units *Taq* DNA polymerase (Bioline, London, UK) and 0.5 µM of each primer. PCR conditions were as follows: 94°C for 3 min, followed by 35 cycles of 45 s at 94°C, 60 s at 45°C (60–64°C for 16S locus), and 60 s at 72°C, followed by 5 min at 72°C. PCR products were purified for sequencing using ExoSAP-IT® (Exonuclease I and Shrimp Alkaline Phosphatase in buffer; USB Corp., Ohio, USA), cleaned with Sephadex®-G50 (GE Healthcare Corp.), and labelled using the BigDye Terminator Sequencing Ready Reaction v3.1 kit (Applied Biosystems). The resulting fragments were separated on an ABI 3130xl genetic analyzer. Sequences were checked, edited, and aligned using Sequencher® v4.5 (Gene Codes Corporation, Ann Arbor, MI, USA). All sequences were deposited in GenBank [accession numbers KF662951–KF663043 and KF725843–KF725869]. Available published sequences of the same gene fragments, to which we could assign a geographic origin, were also included in our phylogenetic analyses [GenBank: DQ401259–DQ401278, GU248382–GU248387, FJ007820–FJ007834, AF202735–AF202753].

## Genetic analyses

Neighbour Joining (NJ) and Maximum Likelihood (ML) phylogenetic trees were inferred for both COI and 16S gene fragments. NJ trees were constructed using evolutionary distances computed using the Maximum Composite Likelihood method and 1000 bootstrap replicate tests in MEGA5 (*Tamura, Peterson & Peterson, 2011*). The best-scoring ML trees for COI were estimated using RAxML-VI-HPC v. 7.2.8 (*Stamatakis, 2006*) on the CIPRES portal at the San Diego Supercomputer Center (http://www.phylo.org), optimising free model parameters and executing 1000 rapid bootstraps. Average genetic distances between the main COI lineages – corrected by the K2P + G substitution model – were carried out using MEGA5. Additionally, to identify lineages in our COI dataset, we used the Automatic Barcode Gap Discovery (ABGD) approach (*Puillandre et al., 2012*) using the webtool (http://wwwabi.snv.jussieu.fr/public/abgd/abgdweb.html).

Intra-population gene diversity *Hs* (standardized haplotype diversity) for COI was computed using the program RAREFAC (*Petit, 1998*) to account for population differences

in sampling size. Nucleotide diversity and pairwise $\Phi_{ST}$ values for COI from all 39 sampled populations were estimated using ARLEQUIN v. 3.1. (*Excoffier, Laval & Schneider, 2005*).

**Testing isolation-by-distance patterns and effect of bird flyways.**
The significance of correlations between pairwise genetic and geographic distances (isolation-by-distance or IBD patterns) was tested using Mantel tests on IBDWS v.3.21 (*Jensen, Bohonak & Kelley, 2005*). Prior to analyses in this section, populations inferred to be introduced intentionally by humans (see Table 1) were removed. For all sampled locations, precise decimal longitude and latitude coordinates were obtained using Google Earth (http://earth.google.com). A geographic distance matrix was then computed using Geographic Distance Matrix Generator v.1.2.3 (Ersts, American Museum of Natural History, Center for Biodiversity and Conservation, http://biodiversityinformatics.amnh.org/open_source/gdmg). We used a population geographic distance matrix (Table S1) and a population genetic distance matrix for COI data ($\Phi_{ST}$ values using Kimura 2-Parameter as the evolutionary model, Table S2). The 99% confidence intervals for the slope and intercept were estimated using Reduced Major Axis (RMA) regression with 30,000 bootstrap randomizations using log km geographical distance.

We used canonical Redundancy Analysis (RDA) in CANOCO (*ter Braak & Šmilauer, 2002*) to estimate the relative contribution of geographic distances, migratory flyways and human introductions on genetic distance between populations. RDA, a multiple linear regression method widely used in community ecology, has recently been applied to infer the role of spatial versus environmental variables in structuring population genetics data (e.g., *Orsini et al., 2013*). As the dependent matrix we used the sample loadings of a Principal Components Analysis calculated on $\Phi_{ST}$ values using Kimura 2-Parameter as the evolutionary model for COI data. Environmental variables were whether the flyway overlapped with the *Artemia* population (0 or 1 depending on the presence of birds from the Atlantic, Central or Pacific flyways in the area) and introduction history (0 or 1). We modelled spatial variables using latitude and longitude ($x$ and $y$).

We used the overlap of sampled populations with the three main migratory flyways in America based on two sources (1) *Boere & Stroud (2006)* for shorebirds and (2) Birdlife International (extracted from http://www.birdlife.org/datazone/userfiles/file/sowb/flyways/). These data were used as a proxy for bird movements between locations. This approach is likely to be a rough approximation to the probability of bird migration between locations, given that more precise estimates cannot be obtained due to the absence of sufficient shorebird ringing data for the whole of the Americas. Even if extensive ringing data was available, this would only estimate current bird movements, whereas climate changes since the Pleistocene are likely to have strongly affected bird movements (*Alerstam, 1993*).

RDA analyses were carried out using each set of migratory flyway data (i.e., *Boere & Stroud (2006)*, and Birdlife International information) and the variance partitioning calculated according to *Borcard, Legendre & Drapeau (1992)* when the model was significant. All environmental variables contributed to the full model, so we constructed
two additional RDAs considering only flyway or introduction history as environmental variables (data files used in RDA are deposited in Dryad DOI http://dx.doi.org/10.5061/dryad.7kb11).

## RESULTS

### Phylogenetic relationships and geographic distribution of lineages

Once PCR primers were removed and sequences trimmed to the same length, the 604 bp COI alignment contained 603 individuals newly sequenced in this study (see Table 1), which collapsed into 93 haplotypes. No ambiguities, insertions, deletions or stop codons were present in the alignment. There were 121 variable sites, 86 of them parsimony informative, and 104 synonymous and 15 non-synonymous substitutions.

Using the default parameters, ABGD did not find any partitions in our dataset, so we reduced X, (i.e., the minimum barcode gap width) as suggested by the program, to 1.0 and used Kimura 80, identifying 12 groups with a prior intraspecific distance of $= 0.0028$ (we use the term 'lineages' thereafter, see below for their geographic distribution). When the prior intraspecific distance increases to 0.0046 (not high by any standards) then the number of partitions reduces again to 1.

Both phylogenetic approaches ML and NJ were highly consistent, with two highly supported main branches displaying a geographic distribution along the continent, one mainly Atlantic (lineages 9–12) and the other split between two sub-branches along the Pacific Rim (lineages 1–7) and in Central Canada (lineage 8). Overall, there were at least 12 mtDNA lineages, most of them well supported (Fig. 2). With the exception of lineage 1, each of the lineages showed a restricted geographic distribution indicating a high level of regional endemism (Fig. 3 for the geographic distribution of the lineages). Lineage 2 was found in a single coastal site in NE Mexico, lineage 3 in the five locations in Peru, lineage 4 in two high altitude populations from Central Mexico, lineages 5 and 7 in Central Chile, lineage 6 in a single sulphate-rich location in NE Mexico, lineage 8 in four locations in Saskatchewan, Canada, lineage 9 in Puerto Rico, lineage 10 in Cape Verde, lineage 11 in several locations around the Caribbean, Baja California and SW Mexico, and lineage 12 in Argentina and Chile. In stark contrast to the rest, lineage 1 was genetically diverse and geographically widespread, found across a large geographical area across both sides of the continent, including Brazil, Chile, Cuba, Jamaica, Mexico, USA and Colombia. Out of the 27 haplotypes in lineage 1, seven haplotypes were detected in SFB and GSL, the two commercialised populations in the USA that have been sources for the invasion in the Mediterranean (J Muñoz, A Gómez, J Figuerola, F Amat, C Rico, AJ Green, 2013 unpublished) (see section *Introduced populations* below). In addition, lineage 1 contained six closely related haplotypes only found in Mono Lake – which harbours a population often considered as a separate species, *A. monica* – and haplotypes found in three Colombian populations (CSC, CT and CGZ), five Mexican populations, and the Jamaican population. Two other lineages were distributed across the continental E-W divide, creating contact zones between lineages. In lineage 12, most of the haplotypes were

Table 2 **Genetic divergence between *Artemia franciscana* mtDNA lineages using COI data.** Genetic Distance K2P + G estimated with MEGA between lineages. Genetic distances higher than (or equal to) 4% (0.04) are marked in bold.

| Lineage (distribution) | 1 | 2 | 3 | 4 | 5 | 6 | 7 | 8 | 9 | 10 | 11 |
|---|---|---|---|---|---|---|---|---|---|---|---|
| 1 (USA + introduced) | - | | | | | | | | | | |
| 2 (NE Mexico) | 0.024 (0.006) | - | | | | | | | | | |
| 3 (Perú) | 0.020 (0.005) | 0.029 (0.007) | - | | | | | | | | |
| 4 (C Mexico) | 0.018 (0.005) | 0.031 (0.007) | 0.027 (0.006) | - | | | | | | | |
| 5 (C Chile) | 0.023 (0.005) | 0.033 (0.008) | 0.031 (0.007) | 0.023 (0.006) | - | | | | | | |
| 6 (NE Mexico) | 0.026 (0.006) | 0.038 (0.008) | 0.038 (0.008) | 0.028 (0.007) | 0.028 (0.007) | - | | | | | |
| 7 (C Chile) | 0.032 (0.007) | **0.043 (0.009)** | 0.032 (0.007) | 0.032 (0.008) | 0.032 (0.007) | 0.039 (0.009) | - | | | | |
| 8 (Canada) | 0.020 (0.005) | 0.034 (0.008) | 0.029 (0.007) | 0.019 (0.005) | 0.025 (0.007) | 0.029 (0.007) | 0.031 (0.008) | - | | | |
| 9 (Puerto Rico) | **0.048 (0.010)** | **0.060 (0.012)** | **0.058 (0.011)** | **0.048 (0.010)** | **0.049 (0.010)** | **0.055 (0.011)** | **0.052 (0.011)** | **0.040 (0.009)** | - | | |
| 10 (Cape Verde) | **0.044 (0.009)** | **0.059 (0.012)** | **0.051 (0.010)** | **0.044 (0.010)** | **0.047 (0.010)** | **0.050 (0.010)** | **0.045 (0.010)** | 0.032 (0.008) | 0.023 (0.006) | - | |
| 11 (Yucatán, Colombia...) | 0.035 (0.008) | **0.044 (0.010)** | **0.043 (0.009)** | 0.033 (0.008) | **0.041 (0.009)** | **0.045 (0.010)** | **0.043 (0.010)** | 0.023 (0.006) | 0.038 (0.008) | 0.034 (0.008) | - |
| 12 (Argentina, Chile) | 0.037 (0.008) | **0.048 (0.010)** | **0.046 (0.009)** | 0.037 (0.008) | **0.043 (0.009)** | **0.047 (0.010)** | **0.044 (0.010)** | 0.025 (0.006) | **0.044 (0.009)** | 0.036 (0.008) | 0.020 (0.005) |

found in Argentinean populations, but four of these haplotypes (three of them shared across populations) were found in a Chilean Altiplano population (Salar de Llamara). Lineage 11, mainly Caribbean, has a few haplotypes in two Mexican populations from the Pacific side (Las Coloradas and Faro San José, where they coexist with lineage 1). Finally, although both lineages 1 and 11 are found in Colombia, they were not found together in any of the populations sampled. The genetic divergence between the 12 main lineages ranged from 1.8% (between lineages 1 and 4) to 6.0% (between lineages 2 and 9) (Table 2).

The 16S alignment contained 408 bp from 122 individuals, which collapsed into 59 haplotypes. There were two singleton indels, 63 variable sites and 43 parsimony informative sites. In contrast to the COI analysis, the NJ and ML reconstructions were poorly resolved, especially the basal branches, but most of the lineages recovered by the COI analyses were also recovered for the 16S data, with variable levels of support (see Fig. 4). COI lineages 3, 5, 7 and 9 were highly supported for both ML and NJ analyses in the 16S analysis, whereas lineages 1 and 2 on the one hand and lineages 4 and 6 on the other, collapsed into poorly supported branches. The 16S analyses allowed us to assign several previously sequenced populations, which we were unable to sample, to COI lineages, particularly in NW America and the Caribbean. In addition, the 16S analysis revealed the presence of two new lineages in the Caribbean, one in the Virgin Islands,

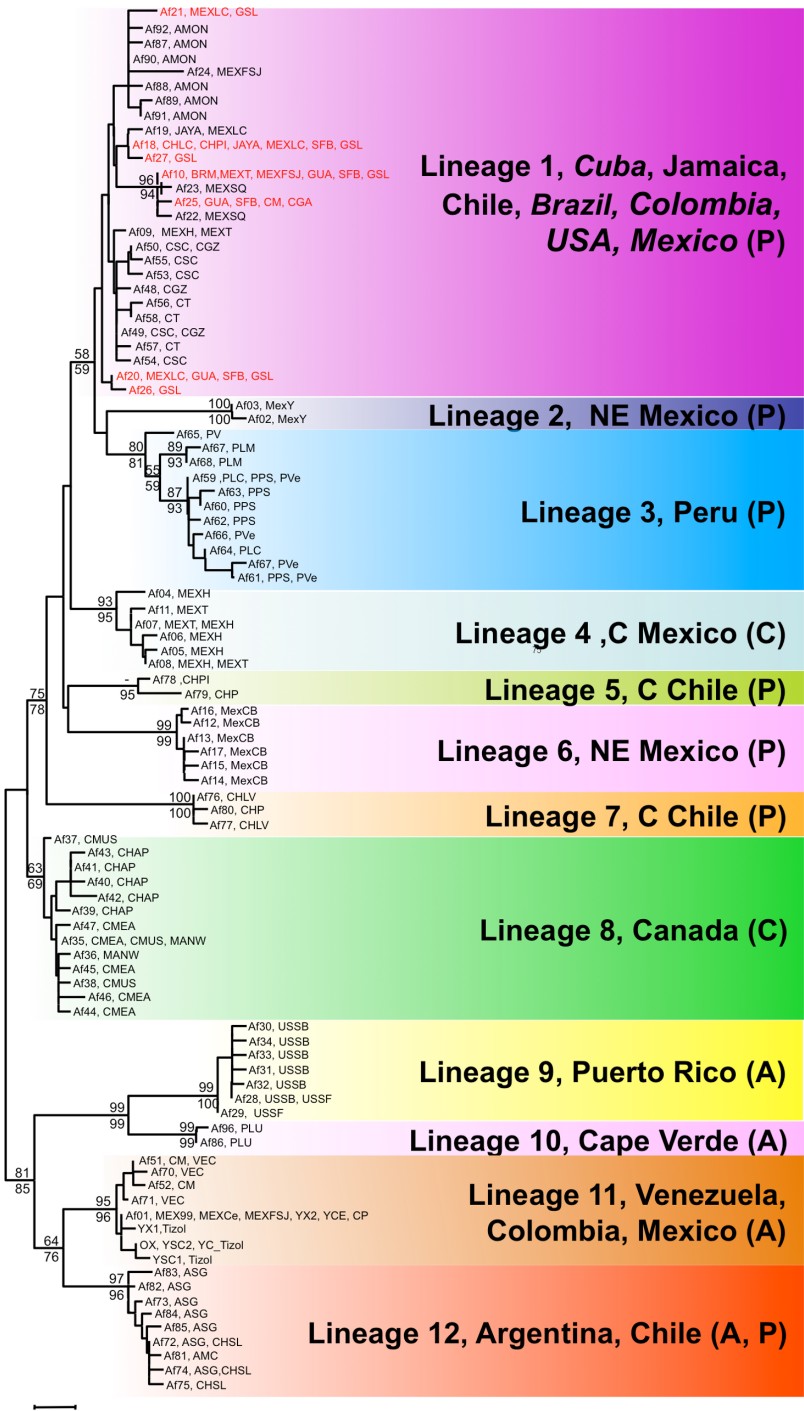

**Figure 2 Phylogenetic relationships of native *Artemia franciscana* COI haplotypes.** The tree topology is the one obtained in the NJ analysis, with bootstrap values shown for NJ (below branches) and ML (above branches). Haplotypes found in the commercialised populations SFB and GSL are marked in red. Haplotype numbers and populations where these were found are noted at the tips. Each lineage label indicates which countries it is found in and its overlap with the Pacific, Atlantic or Central migratory flyways (P, A or C respectively).

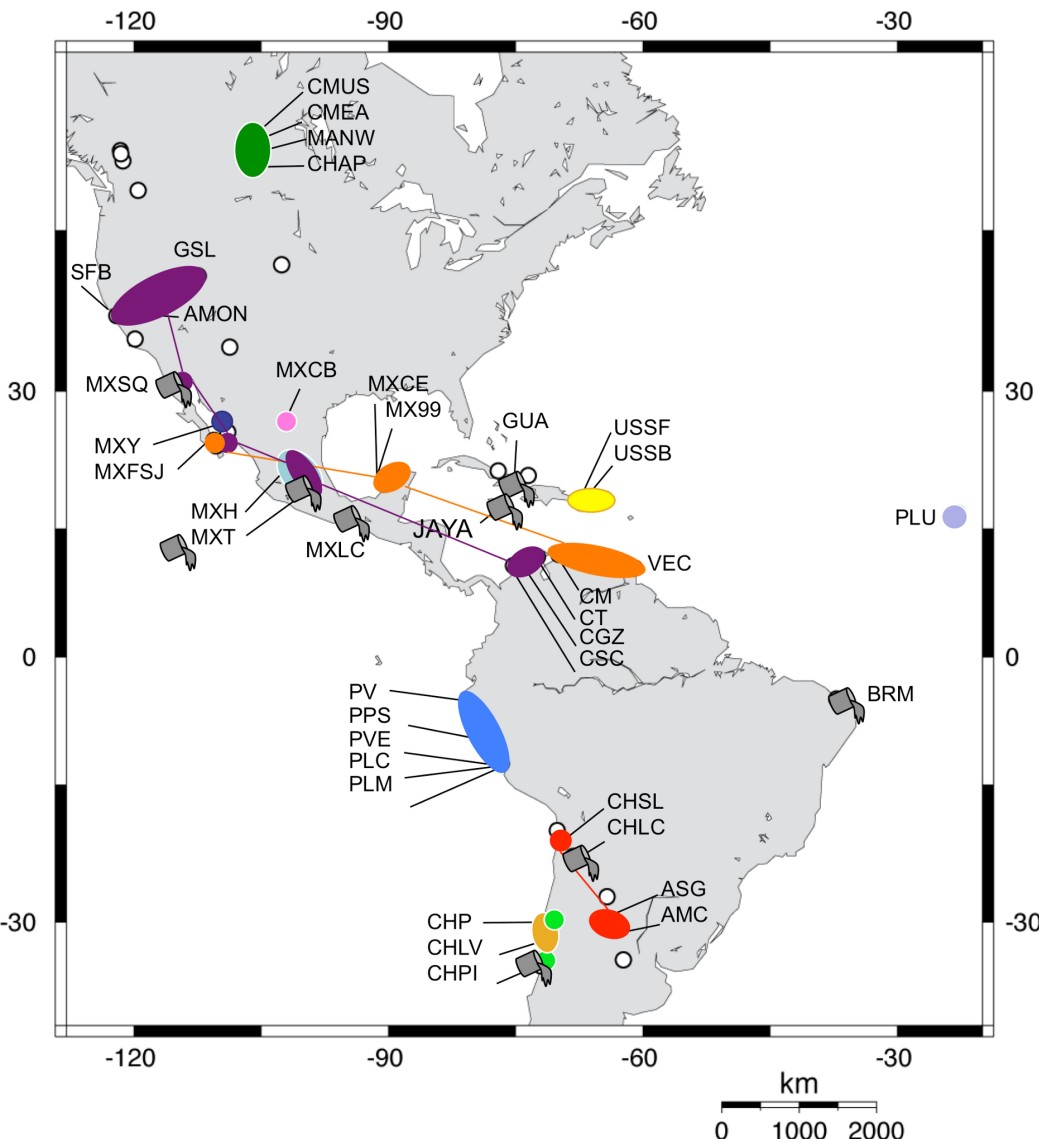

**Figure 3 Geographic distribution of *Artemia franciscana* mtDNA lineages.** The distribution of each COI lineage is shown as areas with the same colour coding as in Fig. 2. Disjunct areas are linked by lines. Introduced populations are denoted by a grey bucket. Only populations sampled for this study are included. Empty circles denote unsampled *A. franciscana* populations.

related to lineage 9 (Puerto Rico) and another in Inague Island (Bahamas) related to lineage 12 (Argentina, Chile). Note that these maintain the relationship with Atlantic lineages. Regarding North-Western American populations – extensively sampled by Prof Sarane Bowen's group – in New Mexico, Nevada and British Columbia, they hold private haplotypes which appear in a poorly supported branch with Mono Lake and Mexican haplotypes. Other populations (Clinton, Basque Lake, Baja California and Carrizo Soda lake) also appear in the composite lineage 1 and 2, underscoring the diversity of USA *A. franciscana* populations. Interestingly, 16S haplotypes from Jesse Lake (Nebraska)

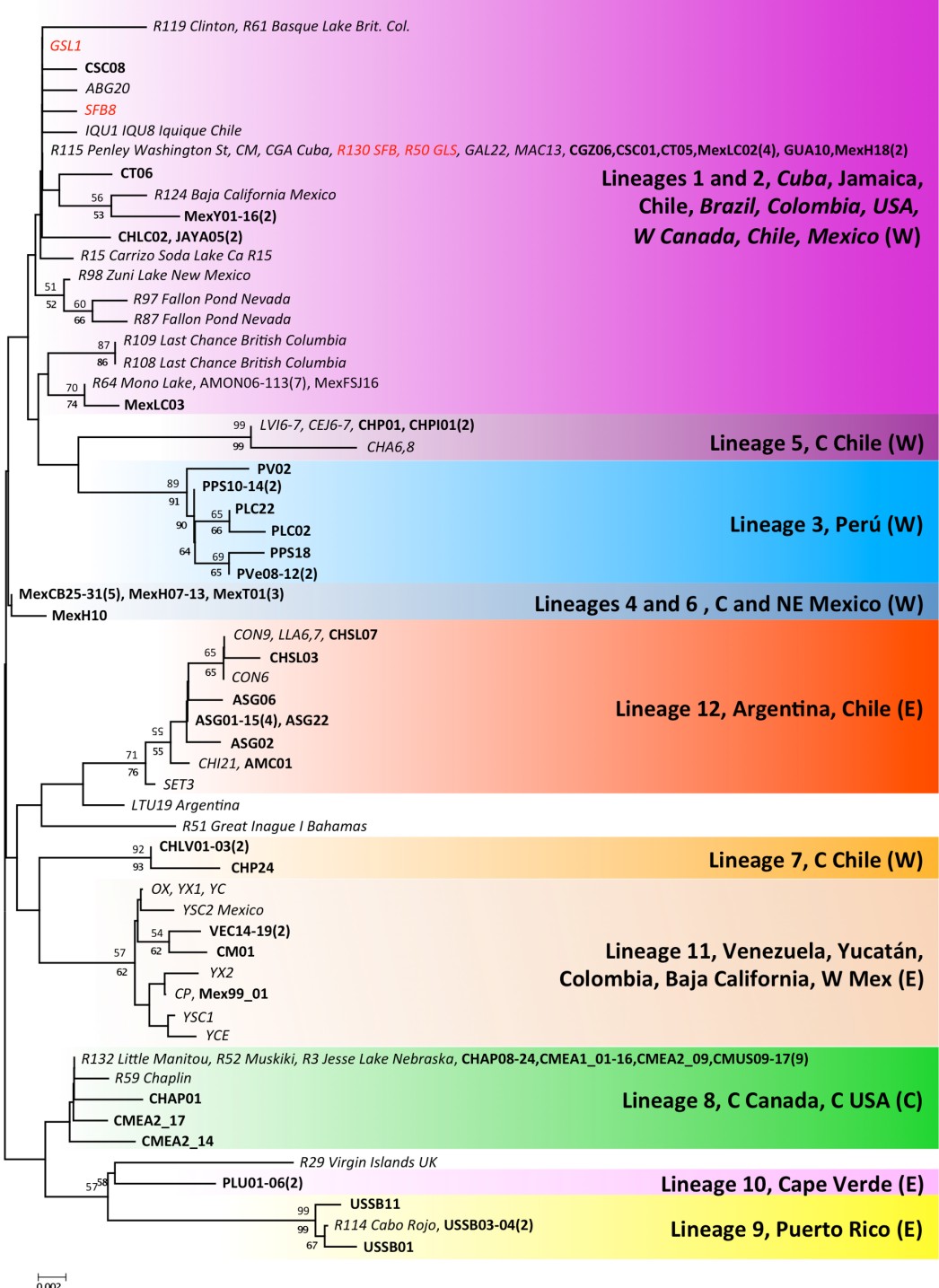

**Figure 4 Phylogenetic relationships for native *Artemia franciscana* 16S haplotypes.** The topology obtained in the NJ analysis is shown, with bootstrap values for NJ (below branches) and ML (above branches). Haplotypes found in the commercialised populations SFB and GSL are marked in red. Bold sequences are those produced in this study and italics those from GenBank. The number of individuals sequenced in each location (unless one) is in parenthesis.

belong to the Central Canadian lineage (lineage 8) together with Little Manitou, Muskiki, Meacham and Chaplin haplotypes.

## Intra- and inter-population genetic diversity in COI

The number of individuals sequenced per population ranged between 4 and 37, depending on available material (average 15.97; see Table 1 for estimates of $\pi$ and $H$, and details of the haplotypes present in each population). The number of haplotypes per population ranged from one to seven. The highest standardized gene diversity ($Hs$) was found in the Mexican population MexH, whereas five populations (MexCe and Mex99 from Mexico, BRM from Brazil, PV from Peru, AMC, from Argentina, and CHLC from Chile) were fixed for a single haplotype. Most haplotypes were found within single countries, except for several haplotypes shared between some countries and the two commercial USA populations SFB and GSL (see Table 1).

Populations were strongly structured genetically (global $\Phi_{ST} = 0.92$; 0.94 when putative introduced populations were removed), with $\Phi_{ST}$ being highly significant between all populations except for three lakes from Central Canada, plus one pair from Chile/Argentina (see Table S2).

## Identification of introduced populations

We found genetic evidence for putative non-native populations originating from the commercialised SFB or GSL in nine sites from Mexico, Cuba, Jamaica, Brazil and Chile. In these populations, at least one sampled individual shared a haplotype with SFB and/or GSL populations (see Table 1). Those populations showed three patterns: (1) all individuals sampled shared haplotypes with SFB and/or GSL (BRM, GUA, and CHLC); (2) populations had haplotypes shared with the commercialised populations and a further haplotype (Af19 for MEXLC and JAYA) which differs from Af18 (a common haplotype in SFB and GSL) by a single substitution; and (3) populations sharing some haplotypes with SFB and/or GSL, but which also had unrelated additional/private haplotypes (MexT, MexSQ, MexFSJ, and CHPI). For four of the nine putative introduced populations, the occurrence of intentional introductions had been previously reported either in the same or nearby sites (see references in Table 1). Note that introduced *A. franciscana* populations are likely to further expand into nearby suitable habitats due to passive dispersal by birds.

## Isolation-by-distance pattern and the role of American migratory flyways

Mantel tests on pairwise genetic and geographic distances for populations ranging from Chile-Argentina to Canada, excluding those inferred to be introduced, revealed a strong IBD pattern (Fig. 5) with a highly significant correlation between pairwise geographic and genetic distances, indicating that geographic distance between populations explains a large proportion of the genetic variability in the sampled area ($R^2$-value $= 0.323$, $p$-value $< 0.001$).

RDA showed that both flyway and introduction history were significantly associated to population genetic distance ($p < 0.02$ for all the correlations with genetic distance,

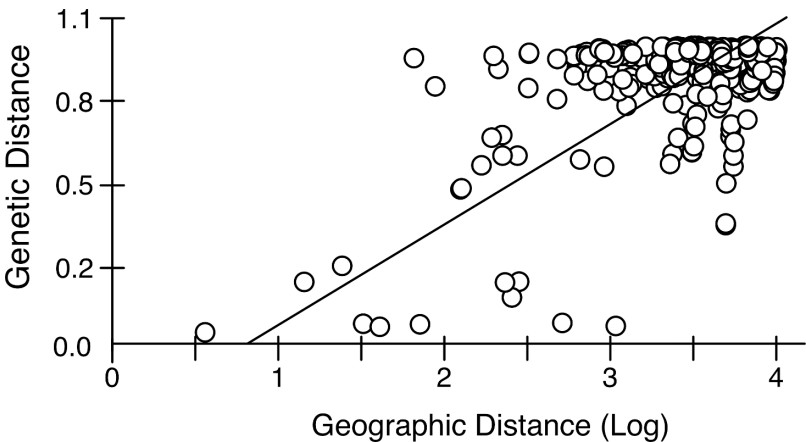

**Figure 5 Isolation by distance in native *Artemia franciscana* populations.** Genetic distance ($\Phi_{ST}$, using Kimura 2-Parameter as evolutionary model, see Table S2) vs. geographical distance (Log geographical distance in km), showing the RMA regression line.

**Table 3 Redundancy analyses (RDA) assessing the contribution of spatial (geographical coordinates of populations) and environmental factors to the genetic distance between *Artemia franciscana* populations.** Explained variance (%) for three RDAs with different environmental variables is given in separate columns. The first RDA included flyway and human introduction as environmental variables, while the others considered only flyway or introduction. Results are given for flyway assignments made according to *Boere & Stroud (2006)* and Birdlife International (http://www.birdlife.org/datazone/userfiles/file/sowb/flyways/) (see Table S2 for matrix details).

| Source of variation | Flyway + introduction | Flyway | Introduction |
|---|---|---|---|
| Space | 8.2/8.7 | 8.3/10.2 | 10.8 |
| Environment | 31.2/30.6 | 21.2/18.7 | 15.6 |
| Environment/Space interaction | 2.4/1.8 | 2.3/0.4 | 0.0 |

whether or not geographical distance was controlled for). The effect of using different flyway data was quite small. Flyway explained 18.7–21.2% of variance in population genetic distance, depending on the flyway dataset used (Table 3). Genetic distance was also affected by historical anthropogenic introductions (15.6% of variance explained) and both factors together (i.e., flyway and anthropogenic introduction) explained 30.6–31.2% of variance in genetic distance. In comparison, geographic distance only explained 8.2–8.7% of variance.

## DISCUSSION

Our analyses revealed that *A. franciscana* has a strong regional genetic structure in its native distribution range throughout the Americas, with twelve largely allopatric endemic lineages. Such high level of population structure, supported by a very high overall $\Phi_{ST}$ value, high number of private haplotypes and significant IBD patterns, indicate that the populations studied are not connected by high ongoing gene flow, pointing instead to the effects of genetic drift and persistent founder effects during historical colonisation

processes and development of local adaptation (i.e., the Monopolisation hypothesis; *De Meester et al., 2002*). The few population pairs with non-significant population differentiation (three lakes from Central Canada, plus one pair from Chile/Argentina) are likely to reflect recent colonisation, rather than ongoing gene flow. Our data also reveals the impact of *A. franciscana* introductions on the phylogeography of the species, as the lineage including the commercialised SFB and GSL populations has now achieved the widest distribution across the continent, in some cases coexisting – and presumably hybridizing – with pre-existing native populations. In addition, our results suggest that migratory birds have an important role in the colonisation of new habitats and are associated with range expansions both in the history of *Artemia* colonisation across the Americas, and also at a local scale, where birds facilitate the expansion of introduced lineages.

## The role of bird migratory flyways

Our study provides new evidence supporting the key historical role of waterbirds as the main factor shaping the population genetic structure of continental aquatic invertebrates at an intra-continental scale. The patterning of the main phylogenetic lineages, with an Atlantic, Central and Pacific distribution – instead of a North American (Nearctic) *vs.* South American (Neotropical) division reflecting the recognized zoogeographic regions and the long isolation of the continents (*Lomolino et al., 2010*; *Holt et al., 2013*) – strongly suggests that historical bird migratory flyways, which occur alongside both the major coasts of this continent, determined the historical spread of *A. franciscana* genetic lineages. Bird movements also might have allowed the subsequent persistence of this structure by facilitating colonisation along each migratory flyway, which shaped the main East-West division in mitochondrial lineages. RDA showed that the effect of migratory flyways was highly significant, and accounted for 20% of the genetic variation between populations once geographic distance was taken into account, suggesting that the distribution of genetic lineages in *A. franciscana* is likely to reflect the impact of historical bird flyways on native phylogeographic patterns. In addition, the strong detected IBD pattern suggests that the chances of bird-mediated colonisation are highly distance dependent (see *Viana et al., 2013*), although instances of long-distance dispersal and colonisation, for example from Argentina to Chile, or to Colombia from northern Caribbean populations, are also apparent from our data. A corollary of our results is that bird movements must have shown some stability, forming parallel N-S flyways during the time frame of *A. franciscana* population diversification (i.e., throughout the Pleistocene) extending into new breeding areas becoming available in the north and new wintering sites in the southern extreme of the continent (*Buehler, Baker & Piersma, 2006*). These results are no surprise given the high transport rates of *Artemia* cysts by waterbirds (*Green et al., 2005*; *Sánchez et al., 2007*) and the lack of other dispersal vectors. Migratory waterbirds have existed since the Early Cretaceous (*Lockley et al., 2012*). Although now-extinct migratory mammals were once major vectors of plant dispersal in the Americas (*Janzen, 1984*), the hypersaline habitats used by *Artemia* are not conducive to dispersal by large mammals.

Our results for *A. franciscana* agree with other studies in North America pointing to an effect of bird movements on the genetic structure of passively dispersed aquatic invertebrates (*Figuerola, Green & Michot, 2005*), but extends these findings to the whole of the Americas, and emphasizes the major role of bird movements in facilitating colonisation into new suitable habitats – in agreement with the patterns found in *A. franciscana* in the invaded Mediterranean range (J Muñoz, A Gómez, J Figuerola, F Amat, C Rico, AJ Green, 2013 unpublished), and those for *A. salina* in its native range (*Muñoz et al., 2008*). Although at a continental scale our results suggest that bird movements do not promote contemporary gene flow between *Artemia* populations, such gene flow may still have a role at a local scale or when new areas suitable for colonization become available.

## Phylogeographic patterns

The high level of endemism and population structure in native *A. franciscana* populations, with low ongoing gene flow and occasional long-distance migration resembles the patterns found in *Artemia salina* (*Muñoz et al., 2008*), a sexual native Mediterranean brine shrimp species, and in a range of other aquatic passively dispersed taxa including sexual reproduction in their life cycles (*Gómez, Carvalho & Lunt, 2000*; *Edmands, 2001*; *De Gelas & De Meester, 2005*; *Mills, Lunt & Gómez, 2007*; *Ketmaier et al., 2008*; *Korn et al., 2010*). Our results also expand and confirm the deep phylogenetic breaks found by *Maniatsi et al. (2009)* in a mtDNA and nDNA study of a more limited number of populations revealing only three lineages. As populations of these passively dispersed organisms can be founded by a small number of propagules, followed by rapid population growth and establishment of large diapausing egg banks, this favours the presence of persistent/long term founder effects, thus reducing the effect of gene flow, possibly reinforced by the development of local adaptation, according to the Monopolisation hypothesis (*De Meester et al., 2002*). Cysts are undoubtedly still regularly dispersed between suitable habitats by waterbirds, but they are unlikely to become established owing to the Monopolisation effects.

Given the range of genetic divergence between lineages (from 2 to 6%) the time frame of their fragmentation can be approximated roughly using a COI molecular clock for other shrimp taxa (1.4% sequence divergence per million year; *Knowlton & Weigt, 1998*), which translates into 1.4–4.3 million years of divergence (reaching the Pliocene), between *A. franciscana* lineages. Even a faster rate of 2% per million years will result on pre-Pleistocene divergence times between the main lineages. Therefore, a contribution of Pliocene/Pleistocene climatic oscillations to population fragmentations after range expansions across the continent can be inferred from our data, possibly allowing survival of lineages in separate geographical areas including Caribbean islands and areas in North and South America. Mexico has the highest lineage richness, with five out of the 12 COI lineages being native to this country – including lineages from both Pacific and Atlantic clades. These findings suggest that this region is likely to have supported separate refugia during climatically adverse periods. The occurrence of a highly divergent Central Canadian prairie lineage was unexpected, as an ice sheet covered this area during the last glacial maximum (*Ehlers & Gibbard, 2004*). However, the 16S data from GenBank suggest that this

lineage also occurs in more southern areas in central USA (Nebraska), where it may have survived south of the ice sheets, and then undergone postglacial colonisation of Central Canada.

Following climate-driven turnover of hypersaline habitats, migratory shorebirds would be involved in expanding the lineages into newly appearing suitable habitats and the chances of successful spread would be strongly distance dependent. However, long-distance colonisation events must have also occurred. For example, assuming that the ancestor of lineages 9 and 10 inhabited the Caribbean area, the colonisation of Argentina by the ancestor of lineage 12 must have entailed successful transfer of some cysts between these distant areas (see Fig. 2). The genetic composition of lineage 12, where most of the haplotypes were found in Argentinean populations (see also Fig. 3) but four of them (three shared across populations) were found in a Chilean Altiplano population (Salar de Llamara), suggests recent colonisation from Argentina to Chile. Finally, the colonisation of Cape Verde Islands, with private haplotypes distantly related to Caribbean lineages, must have involved long-distance transport, possibly from birds accidentally landing there after storms sent them off course, although we cannot rule out the possibility of an unreported human-mediated introduction involving cysts from a Caribbean population not included in our study. Shorebirds are likely vectors for such long-distance dispersal events, and have often been implicated in the dispersal of plants between North and South America, or to oceanic islands (*Cruden, 1966*; *Proctor, 1968*).

The 16S analysis also shows higher richness of lineage 12 in Argentina with two Chilean populations (Convento and Salar de Llamara), for which evidence of nuclear DNA (nDNA) introgression among lineages exists (*Maniatsi et al., 2009*). Furthermore, as migratory flyways overlap on some areas, this could have occasionally resulted in transfers from the Atlantic to the Pacific coasts, as has been suggested for other passively dispersed aquatic species (*Miura et al., 2011*).

Natural spread of lineages from refugial areas is likely to result in contact zones between lineages, which we expect to be sharp, as we found between lineages 1 and 11 in Colombia, where despite two lineages being present in the area, there are no sites where both co-occur.

## Taxonomic considerations

The COI gene is one of the most widely used tools for species delineation (*Hebert et al., 2004*; *Costa et al., 2007*; *Gómez et al., 2007b*). Sequence divergence of 3% have been proposed as a threshold for species delimitation in crustaceans (*Costa et al., 2007*, but see *Lefébure et al., 2006*), but other approaches are also used, such as GMYC (*Pons et al., 2006*) or automatic barcode delimitation, which we used here (*Puillandre et al., 2012*). Our analysis revealed 12 lineages in *A. franciscana*, some of them, like lineages 9 (Puerto Rico) plus 10 (Cape Verde) compared to all the others, or lineages 11 (Circum Caribbean) plus 12 (Argentina-Chile) compared to all the others, have genetic divergences of over 5%. Surprisingly, reproductive isolation – due to ecological specialisation and local adaptation – has only been reported between Mono Lake and San Francisco Bay populations (*Bowen, Buoncristiani & Carl, 1988*) due to the inability of individuals of each of these populations

to survive in each others' ecological conditions, but our data show they are very closely related. Indeed cross-fertility has been observed in the laboratory between the San Francisco Bay population and 15 other populations from the whole range of the species, including some populations included in our study that belong to very divergent mtDNA lineages such as Inague Saltern (Bahamas), Little Manitou (Canada, lineage 8), and Puerto Rico (lineage 9) (*Clark & Bowen, 1976*). Therefore, we concur with *Bowen, Buoncristiani & Carl (1988)* in regarding *A. franciscana* as a very diverse "superspecies", where reproductive isolation mediated by habitat adaptation might occur, but populations in intermediate habitats could act as venues for genetic exchange between ecological isolates.

## Impact of introductions on native population structure and management implications

Given the strong phylogeographic structure of *A. franciscana*, and the high level of private haplotypes found, we used haplotype sharing between the commercialised populations (SFB and GSL) and distant populations as a criterion of recent human mediated introduction. Using this criterion, we identified nine populations where genetic evidence pointed to putative human introductions into Mexico, Cuba, Jamaica, Brazil and Chile. For four of these populations, the occurrence of intentional introductions in the same or nearby sites during the 1970s could be documented. Therefore, our genetic data confirms that the established *A. franciscana* populations in these locations are, at least partially, of introduced origin, and validates our criterion. As for the impact of introductions on native populations, four Mexican populations contained additional private haplotypes not closely related to the introduced ones, which suggests the presence of a pre-existing population before the introduction and the likely introgression of both populations with persistence of native haplotypes, although nuclear loci would be necessary to confirm this. In two populations, we found a discrepancy between our genetic results and previously published ones. The first case is the MEXLC population, which we regard as introduced (lineage 1), whereas in *Tizol-Correa et al. (2009)* the haplotype found is shared with populations from our lineage 11. The second case is CHLC (Laguna Cejas, Chile), which we regard as introduced whereas *Maniatsi et al. (2009)* found that their mtDNA was most closely related to native populations from Central Chile (which belong to our lineage 7). While these authors concluded that the discrepancies between their nDNA and mtDNA data from the latter population pointed to incomplete lineage sorting, an alternative explanation is that they are due to the population being admixed with native and introduced ancestry, as introduced *A. franciscana* is known to hybridize even with the genetically divergent *A. persimilis* (*Kappas et al., 2009*). This failure to detect introduced mtDNA lineages in this population might arise from reduced sample sizes, or indicate real temporal changes in these populations reflecting recent introductions, as samples are likely to have been obtained in different years (unfortunately, no collection dates for these population is reported in *Maniatsi et al., 2009*). Despite human introductions, haplotypes presumably from pre-existent populations have survived and coexist with introduced ones, although a possible loss of genetic diversity due to introductions cannot be ruled out, and should

be investigated in the future, perhaps using sediment cores (a method widely used in the crustacean model organism *Daphnia*) from populations where commercialised strains were introduced.

The high genetic richness found in our COI analyses, and the presence of private haplotypes belonging to lineage 1 in distant populations away from SFB and GSL, including three Colombian and five Mexican populations likely to be native, indicates that the natural distribution of lineage 1 extended further than the two commercialized populations before any human introductions took place. This was also in evidence from our 16S analyses, which included more extended geographic coverage of NW America, and revealed further sites within lineage 1 in the W USA and British Columbia harbouring private haplotypes not found in SFB or GSL. This is also consistent with the presence of private and closely related haplotypes at Mono Lake. The peculiarity and fragmentation of the habitats used by the species, the potential of salinity and varying ionic composition to act as a strong selective agent, the capacity to produce massive quantities of resting eggs that can be readily dispersed by birds, combined with the apparent limitation in modern gene flow, makes this group an ideal system for further studies testing the role of local adaptation and mass effects on reducing gene flow between populations.

Given the impact of the invasive *A. franciscana* across the world, and the high genetic and ecological richness of its native populations, further population translocations should be highly discouraged, and the use of native strains as a source of cysts should be encouraged even within the Americas.

## CONCLUSIONS

Our analyses suggest that *A. franciscana* phylogeography in its native range was shaped by (1) Pliocene/Pleistocene climate fluctuations, which contributed to changes in the areas available to the species, (2) historical bird-mediated colonization along migratory flyways, which shaped the East-West population division, (3) strong and persistent founder events, facilitated by high population growth rates and large population sizes, preventing further gene flow despite ongoing bird-mediated dispersal, and (4) human introductions coupled with regional bird dispersal, explaining the large but localised geographic range of the lineages derived from the commercially exploited North American populations. Our findings suggest that, at a continental scale, bird-mediated transport of invertebrate propagules does not result in substantial ongoing gene flow, but instead determines species phylogeography, facilitating the colonisation of newly available aquatic environments along bird flyways.

## ACKNOWLEDGEMENTS

We thank G Van Stappen (Artemia Reference Centre) for providing the Mono Lake cyst sample and Joachim Mergeay, who read a previous version of this manuscript, and provided many constructive comments including suggesting using RDA. MT Bidwell helped with sampling in Canada.

### Funding

AG was supported by a National Environment Research Council (NERC) Advanced Fellowship (NE/B501298/1). JM was supported by a Junta de Andalucía Excelence Project (P07-RNM-02511) and a European Science Foundation (ESF) grant awarded from the Research Networking Program Activity (ConGenOmics - EX/3646). AJG's sampling work in Canada was supported by a mobility grant from the Spanish Ministry of Science and Innovation (PR2008-0293) and help from MT Bidwell, and was otherwise supported by the Ministerio de Ciencia e Innovación (Project CGL2010-16028, including FEDER funds). The funders had no role in study design, data collection and analysis, decision to publish, or preparation of the manuscript.

### Grant Disclosures

The following grant information was disclosed by the authors:
National Environment Research Council (NERC) Advanced Fellowship: NE/B501298/1.
Junta de Andalucía Excelence Project: P07-RNM-02511.
European Science Foundation (ESF).
Research Networking Program Activity: ConGenOmics - EX/3646.
Spanish Ministry of Science and Innovation: PR2008-0293.
Ministerio de Ciencia e Innovación: Project CGL2010-16028.

### Competing Interests

The authors declare that there are no competing interests.

### Author Contributions

- Joaquín Muñoz conceived and designed the experiments, performed the experiments, analyzed the data, contributed reagents/materials/analysis tools, wrote the paper.
- Francisco Amat and Andy J. Green contributed reagents/materials/analysis tools.
- Jordi Figuerola analyzed the data, contributed reagents/materials/analysis tools.
- Africa Gómez conceived and designed the experiments, analyzed the data, wrote the paper.

### DNA Deposition

The following information was supplied regarding the deposition of DNA sequences:
GenBank Accession numbers for the 16S haplotypes: KF725843-69 and COI KF662951–KF663043.

### Data Deposition

The following information was supplied regarding the deposition of related data:
Dryad DOI: http://dx.doi.org/10.5061/dryad.7kb11.

## Supplemental Information

Supplemental information for this article can be found online at http://dx.doi.org/10.7717/peerj.200.

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
