# Peer review of "Bird migratory flyways influence the phylogeography of the invasive brine shrimp Artemia franciscana in its native American range"

_PeerJ, doi:10.7717/peerj.200_

## Round 0.1 · original submission · Minor Revisions

All reviewers report the manuscript is interesting and provide valuable comments. I suggest the authors carefully address the issues raised by the reviewers, especially those about the experimental design and the validity of your findings (e.g. the explanation of sampling, identification of lineages, alternative explanations besides bird flyways). I have two further comments. 1) The authors are expected to provide the Genbank accession numbers of their sequences when submit the revision. 2) The authors use "Phylogenetic relationships..." and provide the NJ trees in Figure2 and Figure3, but in my opinion, NJ analysis is more like a 'samples assignment' process and it should be careful to say "phylogenetic relationships" based on NJ and without outgroups.

·

Basic reporting

This is a well written manuscript studying the phylogeography of Artemia franciscana in its native range in the Americas (north, central, south). The authors are also interested in seeing the impact of migratory birds and human introductions on the phylogeography.
The background review is sufficient, relevant literature has been cited, and figures/tables relevant.

One suggestion about Figure 1. The migratory flyways as drawn now with the dotted lines looks a bit messy. I would suggest trying some shading (that is slightly transparent). Areas of overlap between the three flyways would automatically be a different color based on the overlap. I would suggest primary colors (red, yellow, blue), so the overlap will be another clear color (i.e. red+yellow=orange).

Experimental design

The authors sequence COI and 16S of individuals from 39 populations and perform NJ and ML phylogenetic analyses. Several statistical analyses are performed to test IBD patterns (Mantel Test) and various population genetic analyses.

One area for improvement is in the explanation of sampling. We know that there are 39 populations, but it is unclear how many individuals per site have been sequenced. A simple range (i.e. between 1-30 individuals were sequenced for each site, determined by the available material). Additionally, it is not clear why the COI dataset (603 individuals) is so much larger than 16s (122 samples). This needs to be explained

Validity of the findings

The authors find that the there is endemism in populations that is not swamped out by bird mediated gene flow and human introductions. This is shown by the presence of private alleles to populations.
Instead, bird mediated gene flow seems to be shaping the phylogeography, as seen by phylogeographic patterns following migration patterns.
This study demonstrates the contribution of birds and humans to the phylogeography of A franciscana.
These conclusions are appropriately stated and backed by data.

Additional comments

I suggest a few small changes that can improve the manuscript
1) Pg 5, Line 71, a comma is missing "American avocet, Recurvirostra americana"
2) The first section in the results "Phylogenetic relationships and geographic distribution of lineages". In this section, there is some very detailed information about the relationship between the localities. This information is important but a bit confusing and difficult to read. Try to simplify it--it will be much easier to read
3) The caption for figure 1 does not make sense. Remove "are shown".
4) I am very interested in seeing the geographic pattern of the lineages in relation to the map. Currently it is difficult to understand. I suggest making another map with pie charts as the locations. The pie charts for each location would be split corresponding to the colors of the lineages (in figure 2). This would make the results of the paper much clearer to understand.

·

Basic reporting

No Comments

Experimental design

clearly explained

Validity of the findings

very good

Additional comments

General comments
1. One weak point is the identification of the 12 lineages from the COI tree. It seems that the lineages have been identified just by eye, without any clear rationale or support. It could be better to use a fixed threshold or the ABGD method or any other tool from DNA taxonomy (e.g.GMYC) to identify lineages. Given that the part describing the lineages is long (line 218-249), a strong rationale to support the reliability of the 12 lineages is pivotal. The same problem exists also for 16S, but it may be considered minor. Moreover, given the discussion on the taxonomic implications (line 425-444), a more quantitative approach in DNA taxonomy should be performed.
2. The inference on bird flyways is in reality based on a East-West division. The East-West division could be due to several alternative explanations, not only to the bird flyway. Unfortunately, none of the alternative explanations is mentioned and only the bird flyway is discussed. It could be nice to include environmental and climatic variables in the analyses, in order to disentangle the role of the habitat and the birds in the East-West division.
3. The fact that the bird flyway had an effect in the past but does not have it now is difficult to reconcile. Why are birds not carrying resting stages over long distances nowadays?
4. The discussion on gene flow is not clear. How was gene flow measured? As the occurrence of COI and 16S haplotypes? This should be clear to all readers.

Minor comments
Line 134: 709bp, but on line 212 it is 604bp. Provide an explanation for the discrepancy.
Line 191: pertenence?
Line 297-299: the reason to support that introduced populations may disperse more than the native ones is obscure. Provide a clear explanation for the statement.
Line 318: why are all the 12 linages considered endemic? Moreover, endemic to what?
Line 319-333: and what about the effect of the monopolisation hypothesis?

---

## Round 0.2 · Minor Revisions

You have basically addressed the issues raised by the reviewers in the revised manuscript. The manuscript will be suitable for publication if it is revised to address the point below.

The authors deposited "All sequences" in GenBank[accession numbers KF662951-KF663043](Line156-157). The accession numbers indicate there are 93 sequences. It seems you only deposited the 93 COI haplotypes (Line228), but failed to deposit the 16S haplotypes (Line143-144) you sequenced. Considering sharing of paper-related data is more and more becoming a condition of manuscript acceptance, you should also deposit your own 16S sequences in GenBank and provide accession numbers in the revised manuscript.

---

## Round 0.3 · accepted · Accept

I also feel sorry for the US government shutdown affecting your sequence submission. I do not want to be stubborn though. So I am happy to accept this interesting manuscript now. Hope the 16S accession numbers can be manually added later.